



# Cloud droplet activation
# in a continental Central European urban environment

Imre SALMA[1], Wanda THÉN[2], Máté VÖRÖSMARTY[2], and András Zénó GYÖNGYÖSI[1]

[1] Institute of Chemistry, Eötvös Loránd University, Budapest, Hungary
[2] Hevesy György Ph. D. School of Chemistry, Eötvös Loránd University, Budapest, Hungary

*Correspondence to*: Imre Salma (salma.imre@ttk.elte.hu)

**Abstract.** Collocated measurements by condensation particle counter, differential mobility particle sizer and cloud condensational nuclei counter instruments were realised in parallel in central Budapest from 15 April 2019 to 14 April 2020 to gain insight into the droplet activation behaviour of urban aerosol particles. The median total particle number concentration was $10.1\times10^3$ cm$^{-3}$. The median concentrations of cloud condensation nuclei (CCN) at water vapour supersaturations ($S$s) of 0.1, 0.2, 0.3, 0.5 and 1.0 % were 0.59, 1.09, 1.39, 1.80 and $2.5\times10^3$ cm$^{-3}$, respectively. They represented from 7 to 27 % of the total particles. The effective critical dry particle diameters ($d_{c,eff}$) were derived utilising the CCN concentrations and particle number size distributions. Their medians were 207, 149, 126, 105 and 80 nm, respectively. They were all positioned within the accumulation mode of the typical particle number size distribution. Their frequency distributions revealed a single peak, which geometric standard deviation increased monotonically with $S$. The broadening indicated larger time variability in the activation properties of smaller particles. The frequency distributions also showed a fine structure. Its several compositional elements seemed to change in a tendentious manner with $S$. They were related to the size-dependent chemical composition and external mixtures of particles. The relationships between the critical $S$ and $d_{c,eff}$ suggested that the urban aerosol particles in Budapest with a diameter larger than approximately 130 nm showed similar hygroscopicity than the continental aerosol in general, while the smaller particles appeared to be less hygroscopic than that. Seasonal cycling of the CCN concentrations and activation fractions implied modest alterations and for the larger $S$s only. They likely reflected the changes in particle number concentrations, chemical composition and mixing state of particles. The seasonal dependencies for $d_{c,eff}$ were featureless, which indicated that the urban particles exhibited more or less similar droplet activation properties over the measurement year. This is different from non-urban locations. The hygroscopicity parameters (κ values) were computed without determining time-dependent chemical composition of particles. Their medians were 0.16, 0.10, 0.07, 0.04 and 0.02, respectively. The averages suggested that the larger particles exhibited considerably higher hygroscopicity than the smaller particles. The urban aerosol was characterised by substantially smaller κ values than for regional or remote locations. All these could be virtually linked to specific source composition in cities. The relatively large variability in the hygroscopicity parameter sets for a given $S$ emphasized that their individual values represented the CCN population in the ambient air, while the averages stood mainly for the particles with a size close to the effective critical dry particle diameters.





## 1 Introduction and objectives

Water is the most abundant vapour in the troposphere. Its condensation onto aerosol particles is the only relevant pathway for cloud or fog droplet formation at water vapour supersaturations ($S$s) occurring in the ambient air (Pruppacher and Klett, 2000). The number and size of the generated droplets depend both on particle properties and local $S$ (Andreae and Rosenfeld, 2008). Only a subset of aerosol particles are able to grow to droplets at a given $S$; they are called cloud condensation nuclei (CCN) for this $S$. From the aerosol side, this ability is primarily controlled by the size, chemical composition and mixing state of particles. The $S$s are mainly governed by cloud dynamics. Different updraft velocities in clouds result in different $S$s, which consequently can change the activation process. As a consequence, the droplet formation can be limited by the presence of CCN and/or by the updraft velocities. The former case, which is called CCN-limited regime, ordinarily prevails in the global troposphere at concentrations of <9000 cm$^{-3}$ (Rosenfeld et al., 2014).

The CCN modify the intensity and other properties of the sunlight reaching the Earth's surface. It is achieved primarily through the droplet number, droplet size and cloud residence time (Andreae and Rosenfeld, 2008; Rosenfeld et al., 2008, 2014). They also influence the hydrological cycle including the precipitation amount and intensity, the vegetation and its interactions with the carbon cycle as well as the atmospheric chemistry, physics and dynamics. Moreover, it is the indirect effects of aerosols via clouds that have the most uncertain contribution to the global radiative forcing calculations (e.g. Carslaw et al., 2013). This is particularly important since the number concentrations of particles seem to be increasing globally due to anthropogenic activities (Andreae et al., 2005). Concentrations of CCN can vary considerably in space and time. Dedicated studies have been performed in field experiments at several locations in the world and at various laboratories (e.g. Dusek et al., 2006; McFiggans et al., 2006; Hudson, 2007; Rose et al., 2008, 2010; Kuwata and Kondo, 2008; Pringle et al., 2010; Wex et al., 2010; Burkart et al., 2011; Sihto et al., 2011; Jurányi et al., 2011; Kerminen et al., 2012; Topping et al., 2012; Paramonov et al., 2015; Herenz et al., 2018; Schmale et al., 2018). Despite their importance, our knowledge on aerosol–water vapour interactions in the atmosphere under $S$ conditions and on cloud microphysics have remained still insufficient. Long-term studies (of at least 1 full year) are required to understand these processes and their consequences. A broad regional coverage is also desired to reach representative results. The data sets for the environmental category of large cities are particularly scarce.

The study presented here deals with the cloud droplet activation of aerosol particles in a continental Central European city, Budapest. It has 2.2 million inhabitants in the metropolitan area, and it is the largest city in the Carpathian Basin. Online aerosol and meteorological measurements have been going on in a semi-continuous manner at the Budapest platform for Aerosol Research and Training (BpART) Laboratory for more than a decennium (Salma et al., 2011; Mikkonen et al., 2020). The related essential





instruments include a differential mobility particle sizer (DMPS) and a condensation particle counter
(CPC). They were complemented by a continuous-flow cloud condensation nuclei counter (CCNc) in
2018. The combinations of the long-term particle number size distributions, total particle number
concentrations and CCN data at various $S$s facilitate the utilisation of special methods for the data
validation and of further joint evaluation procedures.

The major objective of the present study is to gain insight into the droplet activation of urban aerosol
particles based on 1 full measurement year in central Budapest. Specifically: to report, discuss and
interpret the measured time series and descriptive statistics of CCN concentrations and activated fractions
of aerosol particles for various $S$s, to determine the effective activation dry particle diameters, to compute
effective hygroscopicity parameters under supersaturated conditions and to derive some common
consequences of the data sets.
**2 Methods**
The time interval considered in this study was from 15 April 2019 to 14 April 2020. The number of days
with the CPC, DMPS, CCNc and meteorological measurements covered 100, 99, 85 and 100 % of the
relevant days, respectively. The CCNc was out of operation in January 2020. The overall interval also
involved the emergency phase (from 12 to 27 March 2020, 16 d) and the restriction on movement phase
(from 28 March to the end of the measurement year, effectively 18 d) of the first outbreak of the COVID-
19 pandemic in Hungary (Salma et al., 2020b). The clocks of all measuring systems were synchronised
through the computer network with an agreement of ±1 s. Local time (LT=UTC+1 or daylight-saving
time, UTC+2) was chosen as the time base of the data because it was observed that the daily activity time
pattern of inhabitants largely influences many atmospheric processes in cities (Salma et al., 2014;
Mikkonen et al., 2020).
**2.1 Experimental part**
All measurements were performed at the BpART Laboratory (N 47° 28' 30", E 19° 3' 45", 115 m above
mean sea level) of the Eötvös Loránd University (Salma et al., 2016). It represents an average atmospheric
environment for central Budapest due to its geographical location and meteorological conditions. Thus,
it can be regarded as an urban background site. The local sources comprise residential and household
emissions including seasonal heating, exhaust of vehicle traffic and some industrial sources (Salma et al.,
2017, 2020a, 2020b). Long-range transport of air masses can also play a role in shorter time intervals.
The measurement site is located 85 m from the river Danube. The sampling inlets of the instruments were
set up at heights between 12 and 13 m above the street level. Weather shield and insect net were only
adopted to them. The laboratory was air conditioned at 20±3 ºC.





The CPC instrument deployed (TSI, model 3752, USA) was operated with an aerosol inlet flow of 1.5 L
min$^{-1}$, and recorded concentrations of particles with a diameter above 4 nm using n-butanol as a working
fluid. Its sampling inlet was made of stainless-steel tube with a diameter of ¼ inch (6.35 mm) and length
of ca. 1.6 m. Mean particle number concentrations ($N_{CPC}$) with a time resolution of 1 min were extracted
from its extended data base. The nominal specification of the CPC warrants an agreement in
concentrations better than ±10 % between two identical instruments operating in the single-particle
counting mode with a data averaging interval of >30 s.

The DMPS system utilised was a laboratory-made flow-switching-type device (University of Helsinki,
Finland). It measured particle number concentrations in an electrical mobility diameter range from 6 to
1000 nm in the dry state of particles (with a relative humidity of RH<30 %) in 30 channels with a time
resolution of 8 min (Salma et al., 2011, 2016). Its main components included a radioactive ($^{60}$Ni) bipolar
diffusion charger, a monotube Nafion semi-permeable membrane dryer, a 28-cm long Vienna-type
differential mobility analyser and a butanol-based CPC (TSI, model 3775). The aerosol flow in the high
and low modes were 2.0 and 0.31 L min$^{-1}$, respectively. The sheath flows were 10 times larger than the
aerosol flows. The sampling inlet was made of copper tube with a diameter of 6 mm and length of ca. 1.9
m. The measurements were realised semi-continuously according to international technical standards
(Wiedensohler et al., 2012; Schmale et al., 2017).

The CCNc system implemented was a DMT-200 instrument (Droplet Measurement Technologies, USA).
It contains two vertical condensation chambers A and B of cylindrical shape (inner diameter 2.3 cm,
length 50 cm; Roberts and Nenes, 2005; Rose et al., 2008). Their porous internal walls are continuously
wetted with liquid water from peristaltic pumps. A linear positive temperature gradient along the cylinders
is established and controlled at the top, middle and bottom zones of the chambers. The aerosol sample
flow is continuously guided through the centre of the chambers and is surrounded by filtered sheath air
flow. The flows proceed from top to bottom under laminar conditions and near-ambient air pressure ($P$).
As the flows pass through the chambers, heat and water vapor are transported from the internal wall
surface towards the centre of the chambers. Because water molecules diffuse faster than the air molecules
(transferring the heat), a constant water vapor $S$ is generated along the axes. Various $S$s can be adjusted
by selecting different temperature gradients. The particles are exposed to this $S$ for ca. 10 s. Those particles
that activate at a critical $S$ lower than the adjusted value form droplets. Their size is substantially larger
than for inactivated particles. The droplets are detected at the exit of the chambers by optical particle
counters as size distributions in a diameter range from 0.75 to 10 μm. The droplets larger than 1 μm are
considered to be activated CCN, while the concentration of particles in this size interval is negligible.





The total air flow rates were set to 500 cm$^3$ min$^{-1}$ and the ratio of the sample flow rate to the sheath flow
rate was 1:10. The selected $S$s were 0.1, 0.2, 0.3, 0.5 and 1.0 % stepping from the lowest to the highest
values within a measuring cycle with duration times of 12, 5, 5, 5 and 5 min, respectively. The data
measured by the system were recorded every 1 s. The CCN concentrations at a given $S$ ($N_{CCN,S}$) obtained
by the two chambers should not differ by more than 15 %. The system was run in polydisperse operation
mode largely according to the ACTRIS standard operation protocol (Gysel and Stratmann, 2013).

The meteorological measurements took place on site of the BpART Lab. Air temperature ($T$), RH, wind
speed (WS), wind directions (WD), global solar radiation (GRad) and $P$ were obtained by standardised
meteorological sensors (HD52.3D17, Delta OHM, Italy and SMP3 pyranometer, Kipp and Zonen, the
Netherlands, respectively) with a time resolution of 1 min as supporting information.
**2.2 Data treatment and validation**
The measured DMPS data were inverted into discrete size distributions, which were utilised to calculate
particle number concentrations in the diameter ranges from 6 to 25 nm ($N_{6-25}$), from 6 to 100 nm ($N_{6-100}$),
from 30 to 1000 nm ($N_{30-1000}$) and from 6 to 1000 nm ($N_{6-1000}$). The intervals were selected to represent
various important source types of particles and to maintain the comparison with earlier results. The
extraction, treatment and processing of the measured CCNc data including the date and time, $N_{CCN,S}$, flow
rates and activation temperatures ($T_a=(T_1+T_2)/2$, with $T_1$ and $T_2$ as the read wall temperatures at the top
and middle zones of the condensation chambers; Gysel and Stratmann, 2014) were accomplished for each
$S$ stage by a laboratory-developed computer software AeroSoLutions.

The averaging of the individual measured data was performed from the end of each $S$ stage in a backward
direction over a set time span within the temperature stability of the condensation chambers. The
averaging times were determined by examining several randomly selected time-dependencies of CCN
concentrations and temperatures in different seasons. The averaging intervals can be preselected for each
$S$, and were ordinarily set to 90, 210, 210, 180 and 150 s, respectively for $S$s of 0.1, 0.2, 0.3, 0.5 and 1.0
%. Their proper functioning was monitored within the data treatment, and they were refined for some
individual cases if it was necessary. Warning flags on suspicious data were generated in the data
processing, and the filtered data were checked separately. The two data sets for chambers A and B were
averaged if their ratio was <20 %. Otherwise, one of the two data sets was chosen on the basis of their
time evolution. For the $S$ of 0.1 % (for small CCN concentrations), another averaging criterion, namely
ABS($N_{CCN,A}-N_{CCN,B}$)/min(SD$_{CCN,A}$, SD$_{CCN,B}$)<5 was utilised instead of the concentration ratio. The limits
were based upon exercises with concentrations in ordinary measurement intervals. They represent
sensible and pragmatic approaches, although alternative thresholds could also be set. Finally, it was



checked that the averaged CCN concentrations increased monotonically with $S$ within the measurement
cycles. The time resolution of all experimental data derived from the CCNc instrument was 32 min.

The $N_{6-1000}$ data from the DMPS system were compared to the CPC concentrations, which were averaged
over the corresponding DMPS measuring cycle. Due to the differences in the lowest measurable diameters
(6 vs. 4 nm, respectively), an agreement between the two instruments can be expected if the contribution
of nucleation-mode particles to total number of particles is negligible. Additional factor such as larger
particle transport losses along their longer path in the DMPS system and possibly different response times
of the two CPCs involved in the instruments could also add (Salma et al., 2016). The comparison was
realised by evaluating the $N_{CPC}/N_{6-1000}$ ratio as a function of the $N_{6-30}/N_{6-1000}$ ratio. The intercept of their
regression line was considered as the correction factor for the DMPS system (Sect. 3.1).

The CCN concentrations at the $S$ of 1.0 % were compared to the particle numbers. If most particles
activate at this $S$ then the two concentrations are expected to be similar. In a previous survey, certain
criteria were set to exclude the time intervals when very small, hence, non-activating particles were
present in larger concentrations (Schmale at el., 2017). The comparison was performed under the
conditions when the concentration ratio of particles <30 nm to the total particles was <10 % or between
10 and 20 %. These criteria were utilised for remote or regional locations. The conditions seem not to be
applicable for urban data sets since the annual mean and standard deviation (SD) of the $N_{30-1000}/N_{6-1000}$
ratio in Budapest were (52±15) %, and the relative number of DMPS data fulfilling the criterium 1 or 2
above were only 2 % on a yearly scale. This is due to relatively large and persistent contributions of high-
temperature emission sources of particles typically present in cities.

We propose here another criterium for urban or polluted environments so that the $N_{CCN,1.0\%}$ data are
compared to the $N_{30-1000}$ data if the corresponding ratio of $N_{30-1000}/N_{6-1000}$>70 %. This was fulfilled in a
larger number of the DMPS concentrations and yielded more robust statistics. However, the contribution
of smaller particles remained still higher than for the original criteria. The size distribution spectrum
which date and time was the closest (within 20 min) and smaller than or equal to that of the CCN
concentration was considered. The procedure is further discussed in Sect. 3.1.
**2.3 Modelling**
For atmospheric aerosol, the activated fraction of particles tends to increase gradually with the dry particle
diameter (as a sigmoid function instead of a step function valid for internally mixed monodisperse
particles). This is primarily due to the fact that atmospheric particles are often external mixtures, or their
chemical composition changes with particle size (Dusek et al., 2006; Rose et al., 2010). A threshold



activation diameter, which is called effective critical dry particle diameter ($d_{c,eff}$) is defined in these cases
as the size at which 50 % of the dry particles activate at a given $S$ (Rose et al., 2008, 2010).

The effective critical dry particle diameters were assessed from collocated polydisperse CCN and particle
number size distribution measurements as (Sihto et al., 2011; Kerminen et al., 2012; Schmale et al., 2018):
$$N_{CCN,S} = \sum_{i=d_{c,eff}}^{d_{max}} N_i,$$  (1)
where $d_{max}$ is the largest dry particle diameter measured by the sizing instrument (here DMPS) and $N_i$ is
the number of particles in the size channel $i$ of the instrument. Hence, the concentrations were summed
from the largest particle size ($d_{max}$) towards the smaller diameters until the measured CCN concentration
was obtained. In order to estimate the $d_{c,eff}$ with higher accuracy, a logarithmic interpolation was
accomplished between the last 2 diameters of the summation. The size distribution spectrum which date
and time was the closest (with 20 min) and smaller than or equal to that of the CCN concentration was
considered.

It has to be noted that the assumption of internally mixed particles is rarely met in urban environments
including Budapest (Enroth et al., 2018). However, the approximation involves largely compensating
influences, which still lead to reasonable results (Kammermann et al., 2020).

The cloud droplet activation of aerosol particles refers to their indefinite growth (i.e. up to the droplet
sizes) due to condensation of water vapour at constant saturation ratio ($s=p/p_0$ with $p$ being the partial
vapour pressure of water over a droplet solution and $p_0$ being the saturation vapour pressure of water over
pure water with a flat surface). The conditions for $S_{eq}$ (with $S=s–1$) at which the droplets stay in
equilibrium with the water vapour can be described by the Köhler model (e.g. Pruppacher and Klett, 2000;
McFiggans et al., 2006). To calculate the composition-dependent $S_{eq}$ as function of the droplet diameter
($d_{wet}$) for a given dry particle diameter $d_s$, most controlling variables are further simplified and
approximated within different types of thermodynamic parametrisations. In the present study, the
effective hygroscopicity model was adopted (Petters and Kreidenweis, 2007). The $S_{eq}$ can be expressed
in this parametrisation by assuming volume additivity of solute and water in the droplet and spherical
shapes of the dry solute particle and solution droplet as:
$$S_{eq} = \frac{d_{wet}^3 - d_s^3}{d_{wet}^3 - d_s^3(1-\kappa)} \exp\left(\frac{A}{d_{wet}}\right) - 1,$$  (2)
where:
$$A = \frac{4\sigma_{d/a} M_w}{R T \rho_w}.$$  (3)



The $\kappa$, $M_w$, $\rho_w$, $R$ and $T$ are the hygroscopicity parameter, molar mass of water (0.018015 kg mol$^{-1}$),
density of water, the universal gas constant (8.3145 J mol$^{-1}$ K$^{-1}$) and absolute temperature of the droplet
and air in the thermodynamic equilibrium, respectively. The $\kappa$ value expresses the composition dependent
water activity of a solution droplet, thus virtually the influence of the chemical composition of particles
on their CCN activity. Laboratory and field measurements together with modelling considerations have
indicated that this parametrisation proved to be a reliable relationship under both sub- and supersaturation
conditions (Rose et al., 2008; Merikanto et al., 2009; Rissler et al., 2010; Sihto et al., 2011; Kerminen et
al., 2012; Schmale et al., 2018).

The $\sigma_{d/a}$ was assumed to be that of pure water. Some organic chemical species in atmospheric aerosol
particles such as humic-like substances are surface active and can lower the surface tension of the droplets
(Facchini et al., 1999; Ovadnevaite et al., 2017). This depression is mainly controlled by diffusion of
surfactants from the bulk of the droplet to its surface. It takes several hours to reach the thermodynamic
equilibrium at medium concentrations (Salma et al., 2006). This implies that the possible alterations
related to the lower surface tension than for the water are small with respect to estimated experimental
uncertainties and can also be compensated by some surface/bulk partitioning effects (Sorjama et al.,
2004). The surface tension of pure water seems, therefore, to be a reasonable approximation to reality
under the conditions considered in the present study.

The $\kappa$ values can be computed by solving Eq. 2. This contains several independent variables, i.e. $T$, $d_s$
and $d_{wet}$ in addition to the $S_{eq}$ and $\kappa$ value. The $S$s are controlled by the CCNc instrument; the $T$ can be
expressed by the activation temperature in the condensation chamber ($T_a$). For polydisperse atmospheric
aerosol, $d_s$ can be approximated by $d_{c,eff}$ (Eq. 1; Rose et al., 2008). An additional independent relationship,
namely the fact that the dependency of $S_{eq}$ on variable $d_{wet}$ exhibits a maximum (of $S_c$) is also exploited
for solving Eq. 2. The $\kappa$ values were computed in an iterative manner by varying both $\kappa$ and $d_{wet}$ until the
calculated $S$s were equivalent to the adjusted $S$s and at the same time, it showed a maximum (Jurányi et
al., 2010; Rose et al., 2010).

When the volume occupied by the solute can be neglected with respect to the water volume at the stage
of activation, the $S_c$ can be approximated for $\kappa > 0.2$ by (Petters and Kreidenweis, 2007):
$$\ln(S_c) = \sqrt{\frac{4A}{27} \frac{1}{\kappa d_c^3}} \,. \qquad (4)$$

The time resolution of all modelled data was 32 min, which resulted in the total counts of data typically
around $13.6 \times 10^3$ at each $S$ level.





## 3 Results and discussion

The relevant meteorological properties are summarised in Table S1 in the Supplement for the first orientation. They indicate usual weather conditions in Budapest during the measurement year without extraordinary situations.

### 3.1 Data quality

The DMPS measured systematically smaller total particle number concentrations ($N_{6-1000}$) than the CPC ($N_{CPC}$) as discussed in Sect. 2.2. The intercept ($a$) and slope ($b$) with SDs of the regression line of the $N_{CPC}/N_{6-1000}$ ratio vs. $N_{6-30}/N_{6-1000}$ ratio were $a=1.33\pm0.01$ and $b=0.17\pm0.02$, respectively, while the Pearsons's coefficient of correlation ($R$) between the concentration sets was 0.943. As a result of the comparison, a size-independent multiplication correction factor of 1.33 was adopted for the inverted DMPS data.

The scatter plot of the $N_{30-1000}$ DMPS data for which the $N_{30-1000}/N_{6-1000}>70$ % ($N_{30-1000,>70\%}$; Sect. 2.2) and the $N_{CCN,1.0\%}$ data is shown in Fig. 1a. It can be seen that all measured particle number concentrations were larger than for the CCN. The $N_{30-1000,>70\%}/N_{CCN,1.0\%}$ ratio as function of $N_{CCN,1.0\%}$ (Fig. 1b) did not indicate systematic difference between the two instruments. Most concentration ratios with larger (i.e. from ca. 3 to 7) values that showed up in all panels were isolated cases from each other. They were most likely related to the time difference between the actual DMPS and CCNc data. Since the instruments have time resolutions of ca. 8 min and 32 min, respectively, their compared data pairs could have a time difference of up to 16 min (if the first possible DMPS measured spectrum was missing). During this time span, the particle number concentrations could change substantially. This dynamic behaviour is typical for cities and can be witnessed as suddenly appearing stripes on the particle number size distribution surface plots (e.g. Fig. 10 in Salma et al., 2016).

The $N_{30-1000,>70\%}/N_{CCN,1.0\%}$ ratio as function of $N_{30-1000,>70\%}$ (Fig. 1c) suggested that the ratio was slightly increasing with the concentration, mainly above $10^4$ cm$^{-3}$. An agreement between the $N_{30-1000,>70\%}$ and $N_{CCN,1.0\%}$ is expected (approximately within $\pm15$ %; Sect. 2.1) if the number of particles that were >30 nm and that exhibited low hygroscopicity is negligible with respect to $N_{30-1000,>70\%}$. The opposite can be easily realised in cities including Budapest. The argument is backed by the fact that the $R$ between $N_{30-1000,>70\%}$ and $N_{6-25}$ was significant (0.875) during the measurement year. The latter size fraction mainly contains freshly emitted particles from road vehicles in most of the time (Salma et al., 2017), which typically exhibit low hygroscopicity (Burkart et al., 2011; Rose et al., 2011; Enroth et al., 2018). They contribute to the $N_{30-1000,>70\%}$ as well. Another indication of the chemical species with low hygroscopicity is the low contribution of water-soluble organic carbon (WSOC) and high contribution of elemental carbon (soot) to organic carbon (OC), which are roughly related to general hygroscopicity, in central Budapest. The





former ratio was substantially lower (WSOC/OC ratios from 20 to 39 %), while the latter ratio was
considerably larger (EC/OC ratios from 14–20%) in comparison with the aerosol that was already
chemically aged or at regional or remote areas (Salma et al., 2007, 2020a and references therein).


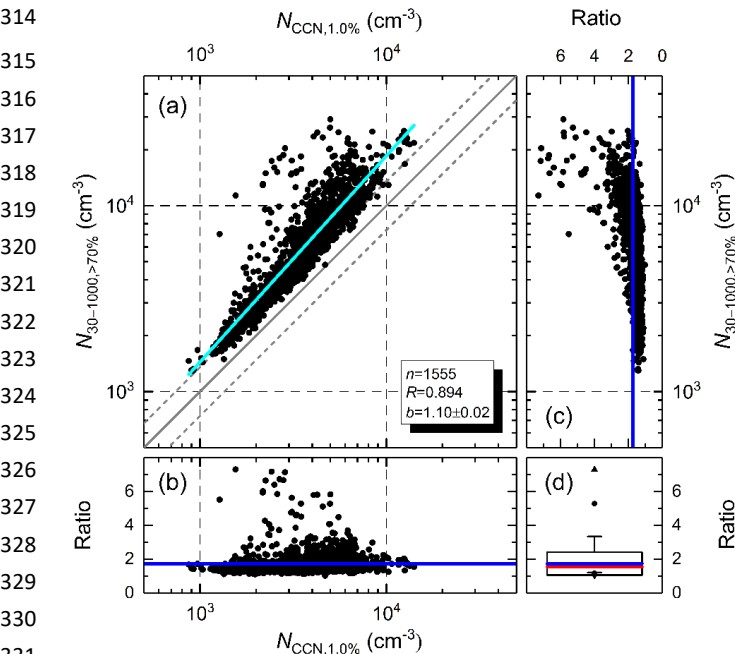

**Figure 1.** Relationship between the concentration of particles with a diameter >30 nm if their relative contribution to the total
particles was >70 % ($N_{30-1000,>70\%}$) and CCN concentration at a supersaturation of 1.0 % ($N_{CCN,1.0\%}$; a). The number of the data
points considered ($n$), their coefficient of correlation ($R$) and the slope ($b$) with SD of the regression line (in cyan) are also
indicated. The line of equality and the dashed grey lines indicate the range of the expected uncertainty of ±15 % solely from
particle counting. The $N_{30-1000,>70\%}/N_{CCN,1.0\%}$ ratios are also shown as function of the variables $N_{CCN,1.0\%}$ (b) and $N_{30-1000,>70\%}$ (c)
with mean (the lines in blue colour). The box and whisker plot summarises the maximum and minimum (triangle pointing
upward and downward, respectively), 1st and 99th percentiles (bullets), mean with SD (blue line and the horizontal borders of
the box, respectively) and median (red line) of the $N_{30-1000,>70\%}/N_{CCN,1.0\%}$ ratios (d).

All this implies that the average $N_{30-1000,>70\%}/N_{CCN,1.0\%}$ ratio has to be extended or even preceded by the
slope of the regression line and $R$ of the two data sets as suggestive metrics for quality assurance. The
mean ratio with SD and median ratio of $N_{30-1000,>70\%}/N_{CCN,1.0\%}$; the slope of the regression line with SD;
and the coefficient of correlation for the overall 1-year-long data set were 1.73±0.67, 1.56, 1.10±0.02 and
0.894, respectively. Our set of quality indicators fits into the results of the data quality check elaborated
for a number of other, mainly regional locations (Schmale et al., 2017). They jointly suggested that the
CCNc and DMPS instruments were operating in a coherent manner and that the CCNc instrument
performed reasonably well over the whole measurement year.



### 3.2 Concentrations and their ratios

The basic statistical measures of the particle number concentrations in different size fractions over the whole measurement year are summarised in Table 1. The mean ratio and SD of $N_{6–100}/N_{6–1000}$ were (81±10) %. The concentrations are comparable with but somewhat larger than our earlier annual results, while the ratios agree well with the previous data (Mikkonen et al., 2020). The median particle number size distribution is shown in Fig. S1.

**Table 1.** Ranges, medians and means with SDs of the particle number concentrations in the diameter ranges from 6 to 25 nm ($N_{6–25}$), from 6 to 100 nm ($N_{6–100}$), from 30 to 1000 nm ($N_{30–1000}$), from 30 to 1000 nm if their contribution to total particles was >70 % ($N_{30–1000,>70\%}$) and from 6 to 1000 nm ($N_{6–1000}$) in a unit of $10^3$ cm$^{-3}$.

| Statistics | $N_{6–25}$ | $N_{6–100}$ | $N_{30–1000}$ | $N_{30–1000,>70\%}$ | $N_{6–1000}$ |
|---|---|---|---|---|---|
| Min | 0.069 | 0.66 | 0.26 | 0.81 | 0.76 |
| Median | 4.0 | 8.2 | 4.9 | 6.0 | 10.1 |
| Max | 137 | 153 | 47 | 39 | 154 |
| Mean | 5.3 | 10.1 | 6.0 | 7.2 | 12.1 |
| SD | 5.1 | 7.4 | 3.9 | 4.6 | 8.1 |

It is noted for completeness that the median $N_{6–100}$ and $N_{6–1000}$ during the restriction on movements of the first outbreak of the COVID-19 pandemic were smaller than the annual median levels by 72 and 79 %, respectively, while the $N_{30–1000}$ remained similar. The mean $N_{6–100}/N_{6–1000}$ and SD of (75±12) % indicated that the share of the ultrafine particles substantially decreased (cf. the previous paragraph). All this is in accordance with the conclusions of our more extensive study dedicated to this issue (Salma et al., 2020b).

The basic statistical measures of the CCN concentrations at different $S$s over the whole measurement year are surveyed in Table 2. It is mentioned for completeness that some individual CCN concentrations at $S$s of 0.5 and 1.0 % were above 9000 cm$^{-3}$, but only in 10 (0.073 % of all relevant data) and 59 cases (0.43 %), respectively, while the related $\kappa$ values were rather low (Sect. 3.5). Therefore, the CCN-limited regime of droplet activation was realised all over the year. The median concentration changed monotonically from $0.59{\times}10^3$ to $2.5{\times}10^3$ cm$^{-3}$ with $S$ and showed a levelling off tendency. They were fitted by a power law function in the form of $N_{CCN,S}=c{\times}S^k$, where $S$ is the supersaturation in % (in Origin Pro 2017 software using the Levenberg–Marquardt algorithm) to obtain the so-called traditional CCN spectrum (Pruppacher and Klett, 2000). The constant $c$ corresponds to the CCN concentration at a $S$ of 1.0 %. The knowledge of these 2 parameters are sufficient for some cloud microphysics applications. The fitted parameters $c$ and $k$ with SDs obtained were $(2.81{\pm}0.12){\times}10^3$ cm$^{-3}$ and 0.52±0.05, respectively. The constant $c$ agreed with the measured average $N_{CCN,1.0\%}$ (Table 2). The exponent $k$ was within the interval reported for other continental locations ($k$=0.4–0.9; Pruppacher and Klett, 2000). The fitted function





reproduced the experimental data at higher ($>$0.2 %) $S$s satisfactorily, while their ratio became 1.25 at a
$S$ of 0.1 %. The comparison of the concentrations with other locations is accomplished together with the
effective critical dry particle diameters in Sect. 3.3.

**Table 2.** Ranges, medians and means with SDs of the CCN concentrations (in $\times 10^3$ cm$^{-3}$) at supersaturations of 0.1, 0.2, 0.3,
0.5 and 1.0 %.

| Statistics | 0.1 % | 0.2 % | 0.3 % | 0.5 % | 1.0 % |
|---|---|---|---|---|---|
| Min | 0.025 | 0.076 | 0.100 | 0.108 | 0.143 |
| Median | 0.59 | 1.09 | 1.39 | 1.80 | 2.5 |
| Max | 2.9 | 5.6 | 8.1 | 10.1 | 14.1 |
| Mean | 0.67 | 1.25 | 1.59 | 2.0 | 2.7 |
| SD | 0.41 | 0.74 | 0.97 | 1.2 | 1.5 |


The mean activation fractions (AF=$N_{CCN,S}/N_{6-1000}$) of the particles increased monotonically from 7 to 27
% with $S$ and showed a levelling off character (Fig. 2). The maximum value was considerably smaller
than for remote or regional locations (Sihto et al, 2011; Paramonov et al., 2015). The shape of the AF
curve was similar to that for the CCN concentrations. This is typical for non-coastal locations, where a
multicomponent mixture of particle sources yield more-or-less balanced and, therefore, similar curves
(Schmale et al., 2018). At the same time, the relative SDs (RSDs) of the mean values were relatively high
(between 45 and 70 %), which pointed to a considerable time variability of both $N_{6-1000}$ and $N_{CCN,S}$. It also
hinted that the prediction of CCN concentrations based solely on particle number concentrations and mean
AFs are expected not to be reliable in urban environments.

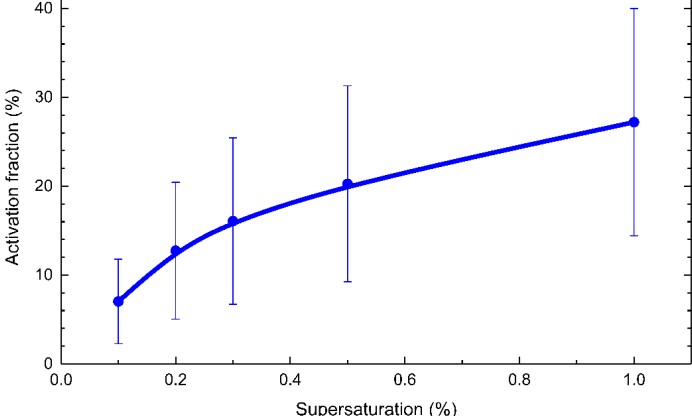

410 **Figure 2.** Mean activated fraction of total particles ($N_{6-1000}$) with SDs at supersaturations of 0.1, 0.2, 0.3, 0.5 and 1.0 %. The

411 line serves to guide the eye.





### 3.3 Effective critical dry particle diameters

Basic statistical measures of the effective critical dry particle diameters at different $S$s over the whole measurement year are displayed in Table 3. The median $d_{c,eff}$ decreased from 207 to 80 nm with $S$. All diameters were positioned within the accumulation mode of the median particle number size distribution (Fig. S1). The monthly mean number median mobility diameters for the Aitken and accumulation modes were typically 26 and 93 nm, respectively with an identical geometric SDs (GSDs) of 2.1 (Salma et al., 2011). The broadening was caused by averaging the individual size distributions. Considering the minimum of the $d_{c,eff}$ data, some individual diameters, in particular for $S$s of 0.5 and 1.0 %, could be shifted to the Aitken mode.

**Table 3.** Ranges, medians and means with SDs of the effective critical dry particle diameters at supersaturations of 0.1, 0.2, 0.3, 0.5 and 1.0 %.

| Statistics | 0.1 % | 0.2 % | 0.3 % | 0.5 % | 1.0 % |
|---|---|---|---|---|---|
| Min | 134 | 92 | 61 | 56 | 30 |
| Median | 207 | 149 | 126 | 105 | 80 |
| Max | 474 | 346 | 271 | 241 | 209 |
| Mean | 213 | 153 | 130 | 109 | 83 |
| SD | 33 | 26 | 24 | 23 | 20 |

The present average diameters and CCN concentrations were larger than for coastal or rural background, forested or remote environments (Henning et al., 2002; Paramonov et al., 2015; Schmale et al., 2018). This confirmed that the water activation properties depend on the aerosol type. Our data were comparable with other urban sites (Kuwata and Kondo, 2008; Rose et al., 2010; Burkart et al., 2011; Meng et al., 2014). The modifications within the location category can likely be associated with relatively large differences between urban aerosol properties. The mean share and SD of ultrafine particles were, for instance, $N_{6–100}/N_{6–1000}=(81\pm10)$ % in Budapest and $N_{13–100}/N_{13–929}=75$ % in Vienna. The present $d_{c,eff}$ data were also contrasted with the computed results for the simulated global continental mean $\kappa$ value and SD of $0.27\pm0.21$ (Pringle et al., 2010) in Fig. 3. The lines were obtained using the parameters given in Sect. 2.3. The data points belong to different parallel lines with a theoretical slope of –3/2. They suggested that the urban aerosol particles in Budapest with a diameter larger than approximately 130 nm showed similar hygroscopicity to continental aerosol in general, while the smaller particles appeared to be less hygroscopic. The distinctions were even larger when European continental aerosol is considered ($\kappa=0.36$; Pringle et al., 2010). The data points fairly tended toward the limiting relationship for insoluble but wettable particles by decreasing diameter. Freshly emitted soot particles could be an example of them (Rose et al., 2011).





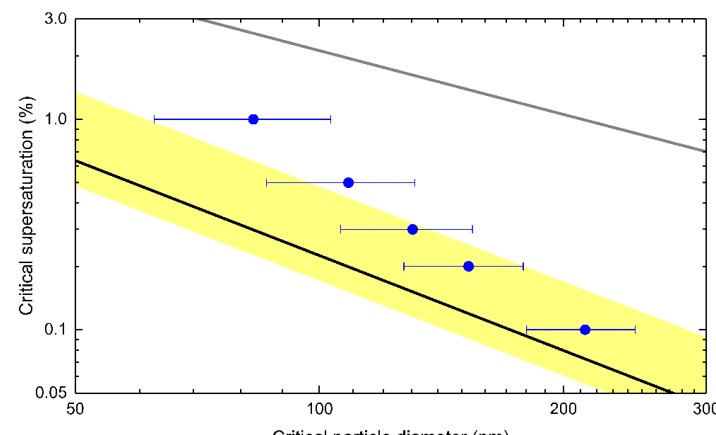

**Figure 3.** Critical supersaturation and effective critical dry particle diameter data pairs with SDs (in blue colour) determined experimentally in central Budapest and the dependency calculated for the simulated global continental mean $\kappa$ and SD of 0.27±0.21. The line in black was obtained for the mean $\kappa$ value, while the yellow bands represent ±1 SD. The relationship for insoluble but wettable particles ($\kappa=0$, the Kelvin term) was displayed by the line in grey for comparative purposes.

The dependency also pointed to the size-dependent chemical composition, which is typical for urban particles. All this is in line with the ideas on the major source types such as vehicle emissions, biomass burning or new particle formation and diameter growth (NPF) events (Salma et al., 2014, 2017, 2020a, 2020b) and the particle number size distributions in Budapest (Salma et al., 2011). Photochemical processing may also play a role through chemical ageing (Furutani et al., 2008).

Frequency distributions of $d_{c,eff}$ at a $S$ can be described by a lognormal distribution function. The normalised differential distributions of the $d_{c,eff}$ data for each $S$ are shown in Fig. 4. They were derived by partitioning all diameter data into 71 intervals with an equal width of 0.0243 on logarithmic scale between 10 and 500 nm. The selections proved to be a reasonable compromise between the good statistics and good data resolution. The distributions exhibited single peaks with geometric SDs increasing monotonically with $S$ as 1.14, 1.16, 1.20, 1.22 and 1.27, respectively. The broadening indicated larger variability in droplet activation properties of the smaller particles.

The peaks exhibited a fine structure. They seemed to contain submodes. This is likely related to the mixtures of particles with different activation properties. The submodes could be produced by sources which result in particles with different chemical composition and mixing states. These differences may not necessarily show up in the particle number size distributions, while they can lead to diverse activation properties. Several compositional elements of the fine structure (e.g. the maximum or the relative peak areas) changed in a tendentious manner by $S$. Their exact identification and interpretation are beyond the objectives of the present paper. They are to be included into an upcoming study which is to deal with the





481  relationships of major source types such as vehicle emissions, NPF events or biomass burning and the

482  activation properties of CCN together with their diurnal variability and air mass trajectories.

483

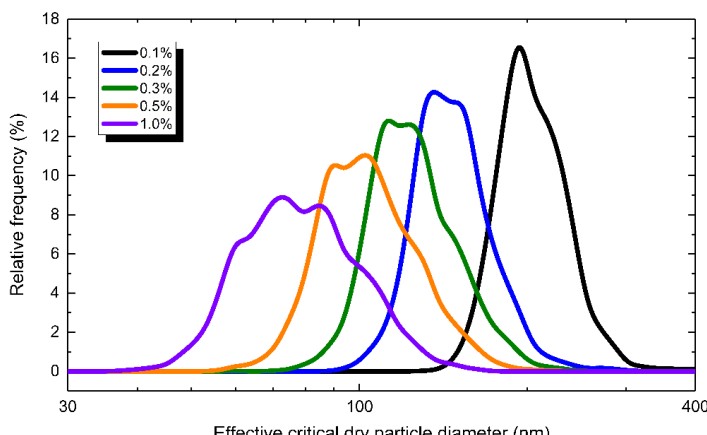

**Figure 4.** Differential frequency distributions of effective critical dry particle diameters at supersaturations of 0.1, 0.2, 0.3, 0.5
and 1.0 % normalised to the total counts of the diameter data.

## 3.4 Seasonal cycling

The time series of the experimental data showed high variability in time. The monthly medians seemed
to be more advantageous for investigating the possible seasonal cycling (Fig. 5). The months were
organised to represent the spring (MAM), summer (JJA), autumn (SON) and winter (DJF) seasons.

The dependencies for the separate variables were similar to each other at different $S$s. The changes were
pronounced mainly for the lager $S$s. The CCN concentrations appeared somewhat smaller from May to
September, and somewhat larger in the other months. Their minimum was typically in May. These
intervals coincided with the non-heating (formally from 15 April to 15 October) and heating seasons (the
rest of the year) in Hungary. As an exception, the concentrations for February were unusually small. The
AFs appeared to be smaller in May (and perhaps also in June) and September (and perhaps also in
October), and larger in the other months. The comparison of the monthly median $N_{\text{CCN},S}$ and AF to the
dependency of the total particle numbers implied that the seasonal variations of the former two properties
were not mainly due to the variations in the particle number concentrations. No obvious dependency for
the monthly median $d_{\text{c,eff}}$ values could be established since their distributions were featureless. The lack
of cycling meant that the particles in Budapest exhibited more or less similar droplet activation behaviour
over the year. This was different from some other, non-urban locations (Pringle et al., 2010; Sihto et al.,
2011; Paramonov et al., 2013, 2015; Schmale et al., 2018). It is noted that the values for 0.1 % $S$ were
somewhat segregated more from the other dependencies.

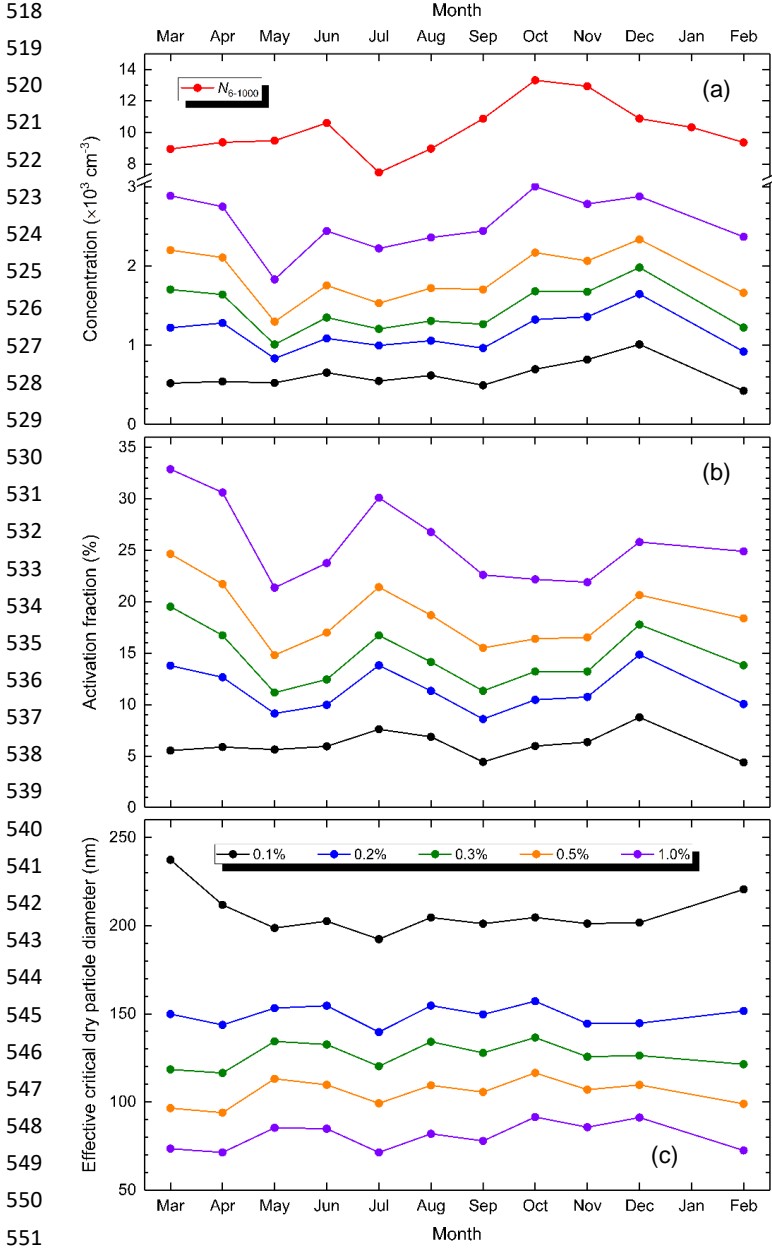

**Figure 5.** Time series of the monthly median CCN concentrations and total particle number concentration ($N_{6-1000}$; a), activation fractions (b) and effective critical dry particle diameters (c) at supersaturations of 0.1, 0.2, 0.3, 0.5 and 1.0 %.

Taking into account that $S$s of only ca. 0.1 % ordinarily occur in warm stratiform clouds, and that these $S$s ordinarily activate larger particles, it can be concluded that the chemical composition of these larger particles were usually balanced over the year. It, therefore, does not seem to play a crucial role in CCN activation even in cities. It has to be immediately added that March and May 2020 were extraordinary due to the first outbreak of the COVID-19 pandemic in Hungary. Firmer seasonal dependencies require



longer continuous measurements since the related properties can be also influenced by inter-annual
differences in chemical composition, aerosol and meteorological properties and in biogenic cycling.

### 3.5 Hygroscopicity parameters

The basic statistical measures of the κ values for different $S$s over the whole measurement year are given
in Table 4. All characteristics decreased monotonically and showed a levelling off tendency with $S$. The
averages implied in general that the larger particles exhibited higher hygroscopicity than the smaller
particles. When considering also the $d_{c,eff}$ data which belong to the $S$s, the present hygroscopicity
parameters fairly agreed with the values derived previously from volatility and hygroscopicity tandem
differential mobility analyser (VH-TDMA) measurements under subsaturated conditions (RH=90 %) at
the identical site (Enroth et al., 2018). In that study, the nearly hydrophobic particles exhibited a mean κ
value of 0.033. The mode typically contained 69 % of particles at a dry diameter of 50 nm, and the κ
seemed to be independent of the particle diameter in the range from 50 to 145 nm. The less hygroscopic
particles showed larger mean κ value of 0.20. They typically made up 59 % of particles at 145 nm.

**Table 4.** Ranges, medians and means with SDs for the hygroscopicity parameter at supersaturations of 0.1, 0.2, 0.3, 0.5 and
575 1.0 %.


| Statistics | 0.1 % | 0.2 % | 0.3 % | 0.5 % | 1.0 % |
|---|---|---|---|---|---|
| Min | 0.014 | 0.010 | 0.006 | 0.003 | 0.0003 |
| Median | 0.159 | 0.099 | 0.071 | 0.043 | 0.021 |
| Max | 0.47 | 0.42 | 0.35 | 0.29 | 0.22 |
| Mean | 0.154 | 0.106 | 0.078 | 0.049 | 0.027 |
| SD | 0.061 | 0.048 | 0.041 | 0.031 | 0.022 |


The average κ values were considerably smaller than for regional or remote locations (Paramonov et al.,
2015; Schmale et al., 2018). There are few hygroscopicity parameters reported specifically for urban
environments, and even less for city centres (Gunthe et al., 2011; Rose et al., 2010, 2011; Meng et al.,
2014; Arub et al., 2020). The present data can also be linked to the average or effective hygroscopicity
parameters found in field measurements and chamber studies for fresh soot particles of <0.01, for
secondary organic aerosol of approximately 0.10 and for inorganic aerosol fraction of ca. 0.64 (Rose et
al., 2011). All this pointed to the presence of freshly emitted and externally mixed soot particles with very
low hygroscopicity in high abundances in central Budapest.

The range of the κ values was increasing with $S$, and, more importantly, it became particularly large (a
factor of ca. $10^3$ for 1.0 %) even when compared with aerosol properties typically driven by atmospheric
dynamics. This can be illustrated by the relationships between the κ value and $d_{c,eff}$ for each $S$ (Fig. 6).

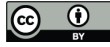



The data sets created separate lines or narrow stripes with a theoretical slope of –3 over the main range
of the variables considered assuming that the other physicochemical properties such as $d_{wet}$, $T_a$, $\sigma_{d/a}$ and
$\rho_w$ do not change substantially. The line for the $S$ of 1.0 % was bended at low $\kappa$ and large $d_{c,eff}$. This is
also in accordance with the $\kappa$–Köhler model (Eq. 2). It can be seen that the data pairs shown for a given
$S$ level indeed cover a wide interval of the related variables. Such variability in $\kappa$ did not back the idea of
using just a single characteristic value for a given $S$, and an effective $\kappa$ parameter or its function on particle
size would be preferred instead (Paramonov et al., 2015). The average hygroscopicity parameter
represented the particles with a size close to the effective critical dry particle diameter. Furthermore, the
distributions of the data pairs along the lines were not completely symmetric with respect to the medians,
which confirmed the possible fine structure of the frequency distributions (Sect. 3.3). The 3 characteristic
points (10th, 50th and 90th percentiles) on the lines indicated a broadening of the frequency distributions
with $S$. The frequency distribution of the hygroscopicity parameters in 71 intervals with an equal width
of 0.0571 on logarithmic scale between $10^{-4}$ and $10^0$ are shown in Fig. S2. They largely reflect the
behaviour and tendencies of the effective critical dry particle diameter (Fig. 4) since their computations
involved $d_{c,eff}$.

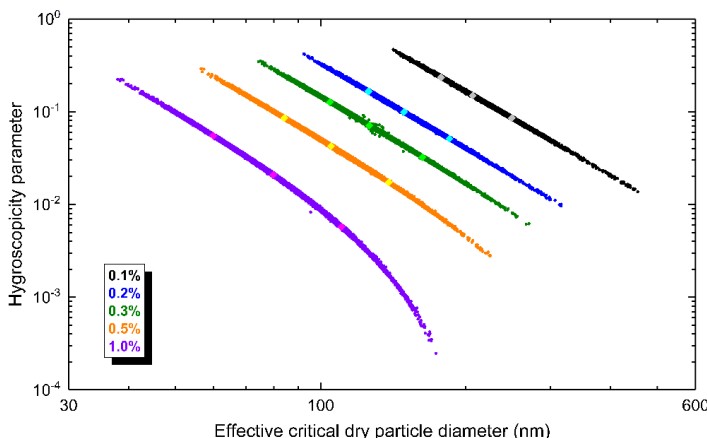

**Figure 6.** Relationship of the hygroscopicity parameter and the effective critical dry particle diameter ($d_{c,eff}$) derived at
supersaturations of 0.1, 0.2, 0.3, 0.5 and 1.0 %. The three diamond symbols in lighter colours on each data set represent the
data pairs belonging to 10th and 90th percentiles; 50th and 50th percentiles; and 90th and 10th percentiles in the order of increasing
$d_{c,eff}$.
**4 Conclusions**
The concentrations of CCN at various $S$s and particle number size distributions were measured in parallel
with each other in a continental Central European urban environment over 1 year. The effective cloud
droplet activation properties of the aerosol population were determined from the available experimental
data without measuring time-resolved chemical composition. The results indicated several urban



specialities. The average CCN concentrations were substantially larger, while the average effective critical dry particle diameters and activation fractions were considerably smaller than for non-urban sites. The particles with a diameter ca. <130 nm already showed lower hygroscopicity than in general, and it was decreasing even more with lower size. The seasonal dependencies of the derived properties were not pronounced or obvious and could not be explained solely by variations in the particle number concentration. These features can likely be related to substantial differences in the size-dependent chemical composition and mixing states of particles, and to the high abundance of freshly emitted less-hygroscopic particles including soot particles in cities. The results and conclusions achieved represent the first information of this type for a city in the Carpathian Basin and contribute to our general knowledge on continental urban atmospheric environments.

The water uptake properties of urban aerosol particles under sub- and supersaturation conditions are increasingly recognised because of their relevance in urban climate considerations and in particle deposition modelling in the human respiratory system. The κ values determined are to be further utilised in health-related studies as well.

After gaining experience with operation and calibration of the dual-chamber CCNc measurement system, we plan to extend one of its chambers by a DMA and CPC setup so we can perform both polydisperse and monodisperse measurements in parallel, which is expected to supply further valuable knowledge on the mixing states of particles. This is especially important since urban aerosol particles typically comprise externally mixed carbonaceous particles with very distinctive hygroscopic properties. This seems to be relevant in general and could also support or facilitate the association of the hygroscopicity parameters to major source types in cities together with multistatistical apportionment methods.

*Data availability.* The observational data are available from the corresponding author.

*Supplement.* The supplement related to this article is available online.

*Author contributions.* IS designed and lead the research; WT and AZGy performed the measurements; MV developed the ASL data evaluation software; IS, MV and WT accomplished the data treatment and prepared the figures; IS, WT, MV and AZGy interpreted the results; IS wrote the manuscript with comments from all coauthors.

*Competing interests.* The authors declare that they have no conflict of interest.

*Financial support.* This research has been supported by the Hungarian Research, Development and Innovation Office (grant no. K132254) and by the European Regional Development Fund and the Hungarian Government (GINOP-2.3.2-15-2016-00028).

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
