# Peer review of "Cloud activation properties of aerosol particles in a continental central European urban environment"

_Atmospheric Chemistry and Physics, 2021_

## Author Comment (AC1)

**Response to Referee number 1**

16th June 2021

The authors would like to thank Referee no. 1 very much for her/his expertise, precise, detailed and very valuable comments to further improve and clarify the MS. We have considered all recommendations and made the appropriate alterations in all cases. We also accomplished some further smaller corrections. Our specific responses are as follows, while the textual modifications amended to the MS can be traced in its marked-up version, which is available online.

**General comments**

The manuscript investigates cloud droplet activation properties of aerosol particles, not really cloud droplet activation (there are studies that differentiate the real cloud droplet population from cloud interstitial particles, and studies that aim to related the cloud droplet population to below-cloud aerosol population) . To avoid any potential confusion, it is important to make this difference in the paper. I therefore strongly suggest modifying the title of the paper into something like "Cloud activation properties of aerosol particles in…". The same concerns wording on lines 9, 67 and 78.

Response 1. The title and related text at the specified locations were changed to conform to the plausible arguments of the Referee.

I appreciate the detailed description of methods used in this investigation. However, there are couple of minor issues related to this section. First, the motivation for the criterion introduced on 198-203 should be improved. What is the real purpose of selecting this ratio, and why to select the value of 70% for this ratio? Second, section 2.3 is not really about modeling, but about using existing mathematical formula. Therefore, the title of that section should be modified into something like "2.3 Calculation of particle hygroscopicity".

Response 2. The main motivation of the comparative exercise, namely to validate the $N_{CCN,1.0\%}$ data by the DMPS measurements was better emphasized. The basic assumption for this validation is that most particles activate at an $S$ of 1.0%. This is evidently not fulfilled over time intervals when the contribution of very small particles (e.g. with diameters <30 nm) is considerable since they do not activate. These data should be excluded from the comparison. The original exclusion criteria – namely that the $N_{6–30}/N_{6–1000}$ ratios are a) to be smaller than 10% or b) to be between 10 and 20% – worked perfectly in remote and regional locations. However, in our case of an urban environment, they jointly resulted in a relative number of only 2% of all DMPS data on a yearly scale. This very small portion is casued by the large contributions of

high-temperature emissions in cities and by new particle formation events, which both result typically in particles with a diameter <30 nm. The former situation often occurs in cities, and, therefore, we had to devote more attention to this issue than at regional or remote locations. We think that the representativity of any conclusion for the whole data set based on such a limited (2%) number of the data could be statistically questioned. Instead, we introduced a new criterium as a compromise between the larger relative number of cases and still constrained contributions of non-activating particles. The limit value of 70% was determined as a compromise in this sense in a pragmatic manner. We are aware that alternative values could have also been set. The paragraph was substantially rearranged and modified at several places to express all this more clearly and directly. As far as the title of the section 2.3 is concerned, we changed it to the formulation suggested by the Referee.

**Detailed comments**

The statement on lines 19-20 (They were related to the size-dependent chemical composition and external mixtures of aerosol particles) gives the wrong impression that this paper measured the aerosol composition and mixing state. It is very true the size-dependent chemical composition and external mixtures of aerosol particles are the most likely explanation for the observations made in this paper. So it should more clearly stated that this is the most likely explanation rather than a real finding of this paper.

Response 3. The sentence under the consideration was removed from the abstract to avoid any possible misunderstanding.

Lines 30-32: Written like it is now, it is not clear what is the result from this particular work (low kappa values in an urban site) and what is based on studies made by others (lower kappa values in regional or remote locations). Please modify.

Response 4. The sentence was reformulated to separate our fidings from the earlier results in an unambigous way.

Lines 43-47: The text is not quite consistent with itself. Since S is affected not only by the updraft velocity but also by sink of water vapor (existing cloud droplet population determined by CCN), I would recommend writing "Different updraft velocities in clouds, together with existing cloud droplet population that depend on CCN concentrations, result in different Ss…".

Response 5. The original statement involved only the main governing property, which is the cloud dynamics, as it was inicated in the sentence. We readily extended the discussion with further possibilities as suggested.

Line 62: This is unclearly written. Maybe one could write "…interactions at S values typical for atmospheric conditions and…"?

Response 6. The suggestion was implemented.

Lines 79-81: I would modify the writing a bit: "Specifically, we will report, … various Ss, in order to determine…".

Response 7. The formulation was modified to: Specifically, we report, discuss, explain and interpret here… .

Lines 426-427: One of the very first studies showing that the minimum diameters of aerosol particles able activate into a cloud droplet is typically well below 100 nm in a remote environment was that of Komppula et al. (2005, J. Geophys. Res., 110, D06204, doi:10.1029/2004JD005200). It might be worth mentioning that paper here.

Response 8. We readily added and cited the reference.

Line 460: Please explicitly write what dependency you refer to here. I assume this refers to the observed slope of the S vs. particle diameter relation in Figure 3 that is different from the theoretical slope for a particle population with a size-independent chemical composition. A reader might not catch this because it requires returning to the information given in the previous paragraph.

Response 9. The text was extended to express explicitly that we discussed here the dependency of the deviation of the experimentaly determined ($d_{c,eff}$, $S_c$) data points from the calculated line for the simulated global continental mean $\kappa$ as a function of $d_{c,eff}$, which was shown in Fig. 3.

Lines 515-516: I do not fully understand this statement. Does it refer to different seasonal behavior of the S=0.1% curve as compared to those of other values of S? The following text (lines 555-558) is also somewhat difficult to understand.

Response 10. The related sentence was reformulated and extended to clarify that the statement refered only to one of the properties in Fig. 5 namely to the $d_{c,eff}$ (Fig. 5c). We also modified and corrected the following two sentences to make them more comprehensible and fluent.

Line 630: ..at lower sizes?

Response 11. The whole sentence was reworded. Firstly, the limiting diameter was specified better as ca.130 nm, secondly and more importantly, we clarified that the rest of the sentence dealt with the difference between the general and urban hygroscopic properties and that this difference increased with decreasing size.

Lines 630-632: I do not understand how particle hygroscopic properties could depend on the particle number concentration. I suppose the authors mean something else here, but it is written in a confusion manner.

Response 12. The sentence was extensively modified to express that we meant the CCN concentrations at various $S$s and activation fractions here.

Table 3: The table caption should explicitly tell that the unit of the numbers given in the table is "nm".

Response 13. The heading of Table 3 was extended by: …in units of nm… .

Imre Salma
corresponding author

---

## Author Comment (AC2)

**Response to Referee number 2**

16th June 2021

The authors would like to thank Referee no. 2 very much for his/her expertise, detailed and very valuable comments to further improve and clarify the MS. We also appreciate his/her ideas on the directions of our potential future research. We have considered all recommendations and made the appropriate alterations. We also accomplished some further smaller corrections. Our specific responses are as follows, while the textual modifications amended to the MS can be traced in its marked-up version, which is available online.

**General comments**

1.  The authors should put their measurements and results in a bigger perspective. The manuscript presents long-term measurements conducted at a single point. How representative are these measurements? Can they tell us anything about the aerosol effects on the cloud formation over urban environments? Considering the size or urban environments compared to, let's say, much bigger marine or forested environments, do we expect any effects of the urban aerosol population and its CCN properties on the actual ambient cloud formation?

    The Conclusions and Sect.3.2 were particularly extended by discussing the spatial representativity of the particular measurement location and the specialities and challenges for the urban-type environment. We also extended the perspective of our results at some other places in the MS.

2.  Conclusions section is fairly short and needs to be expanded. It would be particularly useful to focus on those $S$s found in a typical ambient atmosphere and summarise if and how urban emissions are expected to, at least theoretically, affect the cloud forming potential in urban environments. The authors could elaborate more on how their study compares to similar previously published literature about CCN properties in urban environments and draw conclusions about how their study complemented or added to the existing knowledge. I think it would also be important to notify the reader what else could be done in the future studies to increase the representativeness of single-point measurements and our knowledge of aerosol-cloud interactions.

    This MS presents the first study for Budapest on the hygroscopic properties under supersaturated conditions and is to be followed by further MSs on the time dependencies and source-specific hygroscopicity. Our present knowledge will be hopefully expanded in this field in the future. For the moment, we expanded the MS at several places (e.g. Sect. 3.2) by placing our results into an international frame and by dealing with further possibilities. See also Response to minor comment 11.

**Minor comments**

1.  Introduction – please, give examples of $S$s that can be found in the ambient atmosphere.

    We supplied the typical $S$s in clouds in values.

2.  Lines 42-43 – the particle's ability to act as a CCN is primarily and overwhelmingly controlled by its size and to a much lesser degree by its chemical composition and the mixing state. Please, rephrase.

    The sentence was expanded to express the list/order of importance more specifically.

3.  Line 49 – it's not the CCN, but the droplets that alter the radiation. Aerosol particles also interact with solar radiation, but I think here the discussion is about the droplets.

    The sentence was reworded to emphasize that this is an indirect effect of CCN through cloud droplets.

4.  Lines 63-64 – "Long-term studies (of at least 1 full year) are required to understand [aerosol–water vapour interactions in the atmosphere under S conditions] and their consequences." Has your study done that? I think the statement is rather strong and not supported by your conclusions.

    The statement was refined and softened.

5.  Line 183 – could also be added? Or could also add what?

    The larger diffusion losses in the DMPS with respect to the CPC and the diverse response times of the two instruments could also contribute to the observed differences between the concentrations derived by the DMPS and measured by the CPC. The sentence was broadened to express better our explanation.

6.  Lines 187-203 – these paragraphs are slightly confusing, and I am not sure about their purpose. Why do you want to exclude nucleation mode particles? How does that affect the robustness of your statistics? The measurement location *is* in an urban and polluted environment, so I am not sure why there is a need to define such conditions based on the fraction of nucleation mode particles.

    The main motivation of the comparative exercise, namely to validate the $N_{\text{CCN,1.0\%}}$ data by the DMPS measurements was better emphasized. The basic assumption for this validation is that most particles activate at an $S$ of 1.0%. This is evidently not fulfilled over time intervals when the contribution of very small particles (e.g. with diameters <30 nm) is considerable since they do not activate. These data should be excluded from the comparison. The original exclusion criteria – namely that the $N_{6–30}/N_{6–1000}$ ratios are a) to be smaller than 10% or b) to be between 10 and 20% – worked perfectly in remote and

regional locations. However, in our case of an urban environment, they jointly resulted in a relative number of only 2% of all DMPS data on a yearly scale. This very small portion is casued by the large contributions of high-temperature emissions in cities and by new particle formation events, which both result typically in particles with a diameter <30 nm. The former situation often occurs in cities, and, this was the excess reason why we had to devote more attention to this issue than at regional or remote locations. We think that the representativity of any conclusion for the whole data set based on such a limited (2%) number of the data could be statistically questioned. Instead, we introduced a new criterium as a compromise between the larger relative number of cases and still constrained contributions of non-activating particles. The paragraph was substantially rearranged and modified at several places to express all this more clearly and directly.

7. Line 277 – what is meant by orientation?

We meant gaining the first impression or overview on the actual atmospheric environment. The sentence was modified accordingly.

8. Lines 277-278 – we are not able to see from the table whether there were any extraordinary situations because the table shows values averaged over one year. I think the entire sentence would only make sense if seasonal meteorological values would be shown.

We rearranged the table so that it contains seasonal meteorological mean values.

9. Lines 294-297 – what would cause such rapid concentration changes in an urban environment?

We often experience substantial particle number concentration changes in central Budapest over several tens of minutes. Their actual atmospheric concentrations can vary rapidly because of changes in their important anthropogenic sources in the vicinity. Fluctuating physical removal processes and local meteorological conditions such as WS (influencing their transport) can also play a role. We briefly amended the paragraph by these possible causes.

10. I am not sure if SD is defined anywhere in the manuscript. Is it standard deviation? Or size distribution? What is RSD?

The abbreviation SD was resolved at its first occurrence in the text in line 193 of the discussion paper. Consequently, relative SD (RSD) means relative standard deviation (cf. line 393).

11. Lines 388-411 and Figure 2 – how do the data points and the curve in Figure 2 compare to the curves and the parameterisation presented in Figure 4 of Paramonov et al. (2015)? Lines 394-396 present a very good argument. Is it known whether the considerable time variability of $N_{6-1000}$ is driven by particular size bins? Or does it vary across all size bins? What drives such changes in an urban environment for nucleation mode particles and for those over 100 nm in diameter? This could be of interest since, indeed, both N and CCN seem to show a lot of variability.

We prepared a new Fig. 2 by adding the mean activation curve and its confidence bands from Fig. 4 of Paramonov et al. (2015) for regional and remote locations. We also fitted our experimental data by the same function as described in that paper. All this allowed to interpret our related data more evidently and in a bigger perspective and to formulate our corresponding conclusion explicitly, which all resulted in further improvements of the MS. We appreciate this request. The questions raised later in this comment are also relevant. Temporal variability of particle number concentrations in different size fractions together with CCN concentrations at different $S$ levels have been studied in more detail over various time scales and over certain time intervals that could be assigned to some prevailing source types. The preliminary results have indicated rather complex relationships, which haven't been conclusive yet. We would avoid presenting them here in a touch-and-go manner, and they are to be interpreted more rigorously in a upcoming MS.

12. Line 414 – median $d_c$ decreased with *increasing S*

An increasing, constant or decreasing behaviour of a dependent variable (e.g. function) is interpreted with increasing change of the independent variable – unless otherwise specified. This simplifying convention can substantially shorten the textual formulations of complex relationships and more importantly, it can make their understanding fast and more straightforward. We have used this practice in our previous publications including several ACP articles.

13. Line 428 – "This confirmed that the water activation properties depend on the aerosol type." – yes, this is already well-known. If your "average diameters and CCN concentrations were larger than for" other sites, this means that a) your total particle numbers were higher and b) your particle population was, on the whole, less hygroscopic.

The joint interpretation of our results indeed suggests that both options mentioned by the Referee are valid. Some of the necessary information for this interpretation is discussed and formulated only later in the MS (e.g. Sect. 3.4). Nevertheless, we extended the sentence already here to indicate this additional feature or possibility.

14. Line 431 – the mean fraction

We substituted the word "share" by "contribution".

15. Lines 460-461 – I believe chemical composition is size-dependent for all natural environments, not just urban ones.

We were to state that this dependency is perhaps more pronounced in cities where chemical and physical ageing of particles are limited, and where external mixtures frequently appear or may prevail with regard to regional or remote locations. The sentence was reformulated and extended to cease its ambiguity.

16. Lines 503-504 – "The changes were pronounced mainly for the laRger Ss", which are the least relevant for an ambient atmosphere. This means that these seasonal changes are not very likely to have any effect on the aerosol-cloud interactions.

We readily incorporated this aspect into the text as well.

17. Lines 503-516 – I don't think it is reasonable to talk about monthly changes when there are no error bars with the data points. It doesn't seem like anything is really changing much throughout the year, and there is no need to try and look for such changes (lines 504-509).

The primary message of this part is that there was no obvious cycling of several hygroscopic properties. The interpretation was based on the tendencies in Fig. 5. This shows monthly median values for various variables instead of monthly means and SDs. The selection of medians is justified since these probability variables can be described by lognormal distribution function. At the same time, the error bars for such type of variables (e.g. geometric SDs) are less straightforward. In addition, the error bars are more mandatory when significant tendencies are to be detected and not just lacking dependencies. The corresponding text was modified to indicate this background.

18. Line 558 – should say "March and April" as the campaign ended in April 2020.

The name of the month was corrected.

19. Figure 5 – your measurements were form April 2019 to April 2020. Why is the x-axis from March to February? The only March you measured was in 2020, towards the end of the campaign, but in the figure it appears as the first data point. Please, correct this. Having proper time series shown can also better demonstrate whether any changes occurred during the first COVID outbreak.

We can agree that the suggested ordering of the time scale can be more advantageous when inspecting the effects of the measures due to the first outbreak of the COVID-19 pandemic in Hungary. Our guiding idea in the present paper was, however, to investigate

the seasonal cycling and, therefore, we grouped and organised the months in Fig. 5 from spring (MAM) to winter (DJF) as described in lines 500–501 of the discussion paper. We would like to follow this ordering principle with an added note in the text that the chronological properties and tendencies are to be dealt with in more detail in a separate upcoming MS. See also Response to minor comment no. 11.

20. Lines 558-561 – this could be expanded a bit more. In previous sections you talked about changes in total particle numbers during the first COVID wave. Please, elaborate more here on what happened to $N_{CCN}$, Dc and AF during the last weeks of the campaign.

It is worth noting that the overlapping interval of the measurement campaign and the lockdown pandemic phase (Salma et al.: What can we learn about urban air quality with regard to the first outbreak of the COVID-19 pandemic? A case study from Central Europe, Atmos. Chem. Phys., 20, 15725–15742, 2020) was rather short (18 d) from atmospheric point of view, and this did not allow to arrive at firm conclusions on cloud activation properties of particles. There are a few MSs in preparation within international cooperation, which are expected to supply more information on this issue, but unfortunately, there hasn't been any feasible reference for them yet. The related sentence was extended to indicate this briefly.

21. Line 564 – with *increasing* S

See Response to minor comment no. 12.

Imre Salma
corresponding author